

# Construction of an anthropometric discriminant model for identification of elite swimmers: an adaptive lasso approach

Qile Pan, Rongxin Zhu, Jun Qiu and Guang Cai

Shanghai Research Institute of Sports Science (Shanghai Anti-doping Agency), Shanghai, China

## ABSTRACT

**Background:** Anthropometric characteristics are important factors that affect swimming performance. The aim of this study is to build a discriminant model using anthropometric factors to identify elite short-to-medium-distance freestyle swimmers through an adaptive Lasso approach.

**Methods:** The study recruited 254 swimmers (145 males and 109 females) who were divided them into elite (aged 17.9 ± 2.2 years, FINA points 793.8 ± 73.8) and non-elite (aged 17.1 ± 1.3 years, FINA points 560.6 ± 78.7) groups. Data for 73 variables were obtained, including basic information, anthropometric and derivative indicators. After filtering out highly correlated variables, 24 candidate variables were retained to be used in adaptive Lasso to select variables for prediction of elite swimmers. Deviance and area under the curve (AUC) were applied to assess the goodness of fit and prediction accuracy of the model, respectively.

**Results:** The adaptive Lasso selected 12 variables using the whole sample, with an AUC being 0.926 (95% CI [0.895–0.956]; $P = 2.42 \times 10^{-29}$). In stratified analysis by gender, nine variables were selected for male swimmers with an AUC of 0.921 (95% CI [0.880–0.963]; $P = 8.82 \times 10^{-17}$), and eight variables were for female swimmers with an AUC of 0.941 (95% CI [0.898–0.984]; $P = 7.67 \times 10^{-15}$).

**Conclusion:** The adaptive Lasso showed satisfactory performance in selecting anthropometric characteristics to identify elite swimmers. Additional studies with longitudinal data or data from other ethnicities are needed to validate our findings.

## INTRODUCTION

Anthropometric characteristics are widely considered as important factors affecting sport performance (*Barbosa et al., 2010*; *Saavedra, Escalante & Rodríguez, 2010*; *Ugarkovic et al., 2002*). Anthropometrics that matches the characteristics of a sport helps to master the movement techniques, shorten training time, and plan targeted exercise program to achieve peak performance (*Putra et al., 2020*). In swimming, arm length and girth, or the ratios (*e.g.*, hand-length/forearm-length and forearm-girth/wrist-girth) were found to be significantly associated with the performance of breaststroke (*Sammoud et al., 2018a*), front crawl (*Jürimäe et al., 2007*; *Lätt et al., 2010*) or freestyle (*Nevill, Oxford & Duncan, 2015*), backstroke (*Sammoud et al., 2019*), and butterfly (*Sammoud et al., 2018b*),

Corresponding author
Guang Cai, leng8yang@126.com

highlighting the limb advantage of longer levers. In addition, leg length, trunk breadth (*Sammoud et al., 2018a*), and sitting height (*Sammoud et al., 2019*) were also found to affect swimming performance, but their effect could vary by stroke. From these findings, it is reasonable to hypothesize that anthropometric factors could be useful in the identification of elite swimmers.

Indeed, previous studies have examined the feasibility of identifying elite swimmers by utilizing anthropometric factors. However, in these studies, ordinary regression and discriminant analysis were commonly used to explore the relationship between anthropometric factors and swimming performance (*Lätt et al., 2009a*, *2009b*; *Saavedra, Escalante & Rodríguez, 2010*). The sample sizes are often very limited, with a large number of different anthropometric factors available, and traditional methods often perform poorly. In this study, we tackled the relationship between multiple anthropometric factors and swimming performance using an adaptive Lasso (Least Absolute Shrinkage and Selection Operator) method (*Bhattacharya & McNicholas, 2014*; *Schneider & Wagner, 2012*), which has advantages in consistent variable selection. Thus, the main aim of this study is to build a discriminant model using anthropometric factors to identify elite swimmers through an adaptive Lasso approach.

## MATERIALS AND METHODS

### Participants

From 2016 to 2018, a total of 254 short-to-medium-distance (50-m event, 100-m event, 200-m event, and 400-m event) (*Robertson et al., 2009*) freestyle swimmers were recruited, all of whom were from two provincial teams and three university teams. Swimmers who attained the highest level with a stroke other than freestyle or a distance of more than 400-m were excluded. Skeletal age and menstrual status indicate that these swimmers have largely matured, minimizing the effect of physical maturity on athletic performance (*Vaeyens et al., 2008*).

They were grouped into two categories, *i.e.*, elite (achieving national grade or higher, FINA points: $793.8 \pm 73.8$ at long course 50 ~ 400-m freestyle) and non-elite (achieving $1^{st}$ grade or lower, FINA points: $560.6 \pm 78.7$ at long course 50 ~ 400-m freestyle), according to their highest competitive level which was achieved between 2009 and 2018 and was retrieved from the official website of the General Administration of Sport of China (http://app.gjzwfw.gov.cn/jmopen/webapp/html5/tyzjydyjsdjcx/index.html). The rating standards of the training levels were shown in Table S1.

All the study participants provided written informed consent. Ethical approval of the study was obtained from the Ethics Committee of the School of Life Sciences of Fudan University (No. 497).

### Anthropometry

Somatic measurements were taken by qualified anthropometrists following standardized procedure as specified in the "Shanghai Sport Talent Selection Test Guide" (manual for internal training, File S4). There were 36 assessments: 17 for length, seven for breadth, and 12 for circumference. A second-generation digital display Martin ruler (Martin-CHN;

Shanghai Quyi Technology Co., Ltd, Shanghai, China) and a standard soft tape measure were used. The intra-assessment agreement (technical error of measurement, TEM) of these measurements was 0.26-cm (95% CI [0.20–0.32]) (*Hardy et al., 2018*). Body composition data were obtained by a multi-frequency body composition analyser (Inbody 720; Biospace, Seoul, Korea).

Considering that season had minimal effect on our measurements (*Putra et al., 2020*), and the cooperativeness of the athletes and coaches, each swimmer was measured only once in December (*i.e.*, off-season) during the study. Measurements were taken in the morning, starting with a body composition test in a fasting situation. The rest of the anthropometric measurements were conducted half an hour after eating. Of the swimmers tested during a day, we randomly selected 3% for re-testing to verify reliability. Other basic information of the swimmers was collected from the test record form.

In addition, 28 derivative indexes were calculated, and basic information, such as age of attending the highest level, gender, and competitive level, were also collected, leading to a total of 73 characteristic variables (Table S2).

## Statistical analysis

Among the 73 variables in this study, competitive level (elite *vs* non-elite) is a binary dependent variable, gender is a binary predictor, and the remaining 71 are continuous variables. The pairwise correlation between the variables were calculated to eliminate variables that were highly correlated (*i.e.*, the absolute value of the correlation coefficient |r| >0.7 in the corresponding column).

The adaptive Lasso-penalized logistic model was then adopted to identify optimal anthropometric variables for the identification of elite swimmers. Although this method may produce some degree of asymptotic bias for the non-zero regression parameters, it can provide consistent variable selection and is efficient in coefficient estimation (*Das, Gregory & Lahiri, 2019*; *Zheng, Gallagher & Kulasekera, 2016*). Moreover, it can reduce estimation bias and improve prediction accuracy (*Alhamzawi & Ali, 2018*). The model takes the following form:

$$logit(\pi_i) = ln\left(\frac{\pi_i}{1 - \pi_i}\right) = X^T\beta,$$

where $\pi_i = P(y_i = 1|x_i)$ represents the probability of being an elite swimmer, given the anthropometric characteristics, X is a n × p design matrix, and $\beta$ is the corresponding coefficient vector of p × 1. $\beta$ can be estimated using adaptive penalty by:

$$\beta = \arg_\beta \min\left\{-\sum_{i=1}^{n}[y_i \ln(\pi_i) + (1 - y_i)\ln(1 - \pi_i)] + \lambda \sum_{i=1}^{\rho}\omega_i|\beta_i|\right\},$$

where the penalty parameter λ is commonly determined using the k-fold cross-validation or the Bayesian information criterion (BIC) (*Bhattacharya & McNicholas, 2014*; *Schwarz, 1978*), and the penalty weights $\omega_i$ is generally taken as the inverse of the absolute value of the estimated coefficients from the ordinary least square (OLS) or ridge regression (*Alhamzawi & Ali, 2018*). In this study, we used 10-fold cross-validation to determine λ

**Table 1 Basic information of the included swimmers.**

| Variables | Elite swimmers ($n$ = 89) | Non-elite swimmers ($n$ = 165) | $P$ |
|---|---|---|---|
| Age of attaining the highest level (years) | 17.94 ± 2.17 | 17.11 ± 1.28 | 0.001 |
| Gender, female ($n$, %) | 41 (46%) | 68 (41%) | 0.5 |
| FINA point | 793.8 ± 73.8 | 560.6 ± 78.7 | <0.001 |
| Stature (cm) | 179 ± 8 | 176 ± 7 | <0.001 |
| Weight (kg) | 72 ± 11 | 70 ± 11 | 0.2 |
| Back of foot height (cm) | 7.58 ± 0.53 | 7.50 ± 0.58 | 0.3 |
| Head circumference (cm) | 56.47 ± 1.85 | 57.14 ± 1.60 | 0.004 |
| Body fat percentage (%) | 15 ± 6 | 18 ± 7 | 0.007 |
| Skeletal muscle mass (kg) | 34.5 ± 6.4 | 32.3 ± 6.3 | 0.009 |
| Body mass index (BMI, kg/m$^2$) | 22.23 ± 2.15 | 22.50 ± 2.80 | 0.4 |
| Ratio of arm to stature | 44.18 ± 0.84 | 43.80 ± 0.96 | 0.001 |
| Ratio of arm to leg length A | 81.70 ± 1.53 | 81.18 ± 1.84 | 0.018 |
| Torso length (cm) | 54.6 ± 3.2 | 52.4 ± 3.0 | <0.001 |
| Torso length index | 1.92 ± 0.11 | 1.86 ± 0.11 | <0.001 |
| Biceps circumference difference (cm) | 2.15 ± 0.85 | 1.84 ± 0.80 | 0.004 |
| Ratio of hand breadth to length | 43.31 ± 1.93 | 43.75 ± 2.13 | 0.093 |
| Ratio of ankle circumference to achillis tendon length | 96 ± 8 | 100 ± 9 | 0.003 |
| Ratio of upper to lower leg length | 115.3 ± 3.7 | 115.5 ± 3.9 | 0.8 |
| Ratio of leg length A to stature | 54.09 ± 1.04 | 53.97 ± 1.11 | 0.4 |
| Leg length A to H | 93.09 ± 1.00 | 92.67 ± 0.95 | 0.001 |
| Leg length B to H | 87.47 ± 0.88 | 87.37 ± 1.12 | 0.4 |
| Leg length C to H | 76.50 ± 1.26 | 75.95 ± 1.26 | 0.001 |
| Ratio of bitrochanteric to biacromial breadth | 76.4 ± 3.6 | 77.9 ± 4.1 | 0.003 |
| Ratio of biiliocristal to bitrochanteric breadth | 86.9 ± 2.9 | 86.6 ± 3.3 | 0.5 |
| Ratio of waist to chest | 79.0 ± 3.3 | 81.2 ± 4.5 | <0.001 |
| Ratio of waist to hip | 83.0 ± 3.3 | 82.3 ± 4.4 | 0.2 |
| Ratio of foot breadth to length | 36.08 ± 1.75 | 36.29 ± 1.68 | 0.3 |

Notes:
Data were presented as mean ± SD or $n$ (%). The $P$-values were calculated by Welch two sample t-test and Pearson's Chi-squared test.
SD, standard deviation.
BMI means weight/height$^2$ (kg/m$^2$); Arm means upper limb length (cm); Leg length A means iliospinale posterior height (cm); Torso length means acromion height—trochanterion height (cm); Torso length index means torso length/biiliocristal breadth; Biceps circumference difference means maximum biceps circumference—minimum biceps circumference (cm); Upper leg length means thigh length (cm); Leg length H means eristailiaca height (cm); Leg length B means trochanterion height (cm); Leg length C means gluteal height (cm).

and OLS to determine $\omega_i$. The goodness of fit and prediction accuracy of the constructed model was assessed by deviance and area under the curve (AUC), respectively.

## RESULTS

Table 1 illustrates the basic information of the swimmers. A total of 71 continuous variables composed a 71 × 71 lower triangular matrix with the diagonal and the above elements being zero. The pairwise correlation analysis eliminated 48 continuous variables, leading to a total of 24 candidate variables (23 continuous plus gender) for subsequent analyses (Table S3).

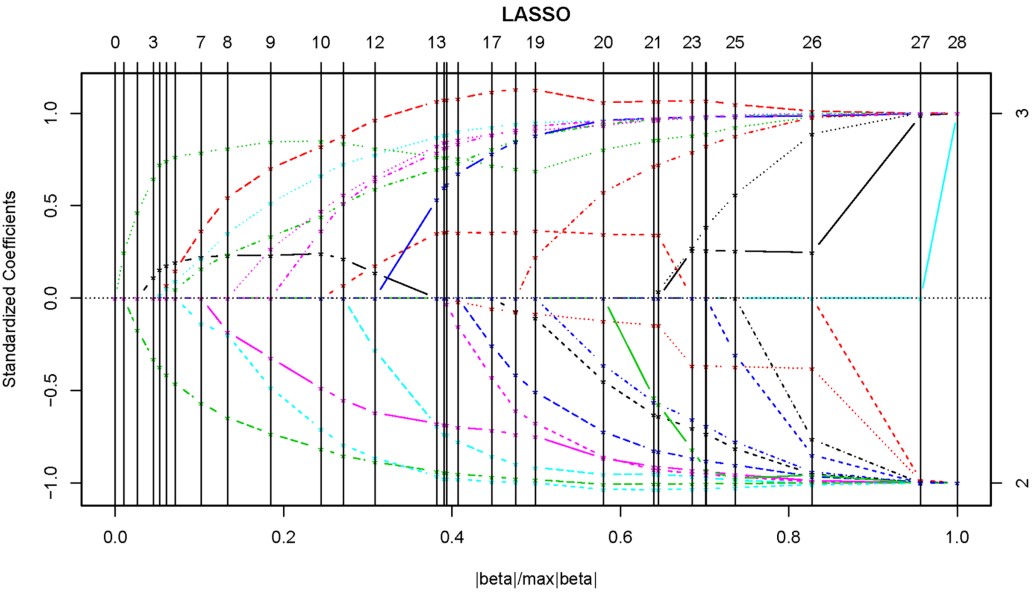

**Figure 1 The coefficient solution path of adaptive Lasso.**

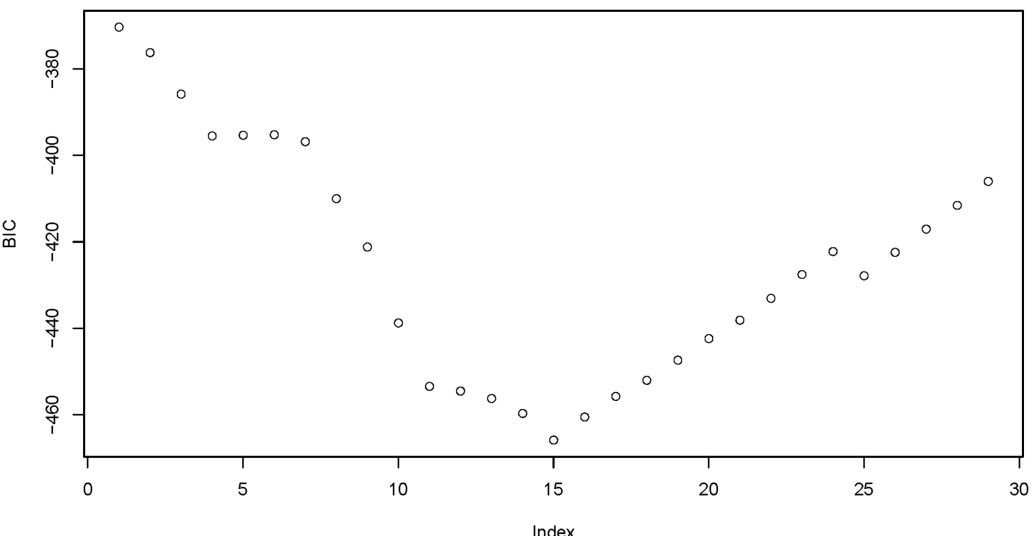

**Figure 2 BIC variation with the number of entered variables.** BIC, the Bayesian Information Criterion. The minimum BIC was −465.387 when 15 variables entered.

In the whole-sample model, *gender* was firstly converted into a dummy variable, with 0 being male and 1 being female. The obtained coefficient solution path and the corresponding BIC were shown in Figs. 1 and 2, respectively, and the minimum BIC was −465.387. The penalty parameter was then chosen to shrink the coefficients of some of the candidate variables to zero (*i.e.*, excluded from the model), and finally 12 out of the 24 variables were selected (Table 2).

**Table 2  Logistic regression results using variables selected by adaptive Lasso (whole sample).**

|  | Coefficient | OR | 95% CI | P |
|---|---|---|---|---|
| Intercept | −69.203 | – | – | 0.014 |
| Age of attaining the highest level | 0.304 | 1.355 | [1.052–1.789] | 0.024 |
| Gender, male | 5.301 | 200.591 | [32.352–1,577.368] | <0.0001 |
| Ratio of arm to leg length A | 0.190 | 1.209 | [0.942–1.566] | 0.141 |
| Ratio of hand breadth to length | −0.153 | 0.859 | [0.683–1.064] | 0.174 |
| Torso length | 0.294 | 1.342 | [1.148–1.585] | <0.0001 |
| Leg length A to H | 0.942 | 2.566 | [1.601–4.313] | <0.0001 |
| Ratio of bitrochanteric to biacromial breadth | −0.201 | 0.818 | [0.698–0.947] | 0.009 |
| Head circumference | −0.787 | 0.455 | [0.32–0.624] | <0.0001 |
| Ratio of waist to chest | −0.264 | 0.768 | [0.647–0.897] | 0.001 |
| Ratio of waist to hip | 0.239 | 1.27 | [1.107–1.474] | 0.001 |
| Biceps circumference difference | 0.407 | 1.502 | [0.86–2.692] | 0.159 |
| Skeletal muscle mass | 0.287 | 1.332 | [1.165–1.542] | <0.0001 |

Notes:
The penalty parameter was then chosen to shrink the coefficients of some of the candidate variables to zero (*i.e.*, excluded from the model), and finally 12 out of the 24 variables were selected for the whole sample.
OR, odds ratio; CI, confidence interval.
Arm means upper limb length (cm); Leg length A means iliospinale posterior height (cm); Torso length means acromion height—trochanterion height (cm); Leg length H means eristailiaca height (cm); Biceps circumference difference means maximum biceps circumference—minimum biceps circumference (cm).

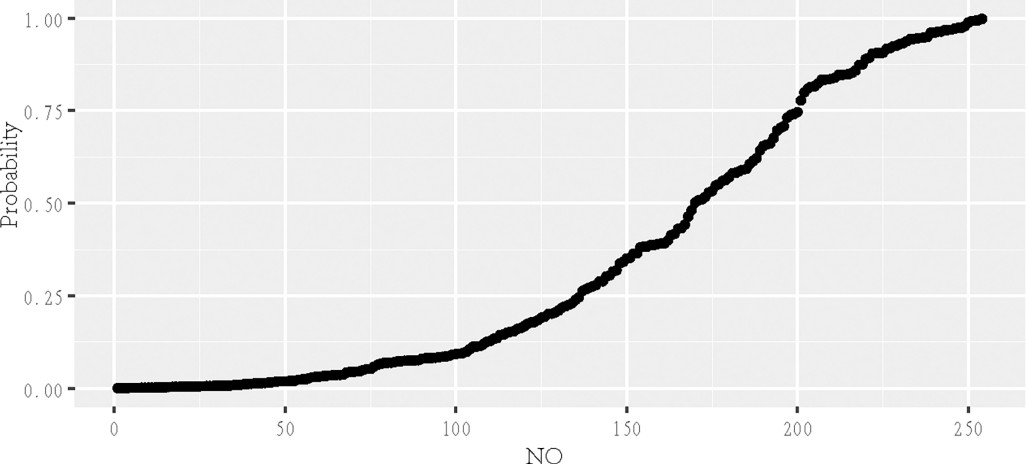

**Figure 3  Fitted logistic graph of the probability of being an elite swimmer.** The predicted values were between 0 and 1, representing the probability of becoming an elite athlete.

According to the chosen variables, a logistic model was built and statistical inference was performed (Fig. 3). The predicted values were between 0 and 1, representing the probability of becoming an elite athlete. The estimated coefficients of the selected variables were shown in Table 2. This model explained 48.23% of the deviance. The receiver operating characteristics (ROC) curve was shown in Fig. 4, with an AUC (obtained by the bootstrapping) of 0.926 (95% CI [0.895–0.956]; $P = 2.42 \times 10^{-29}$).

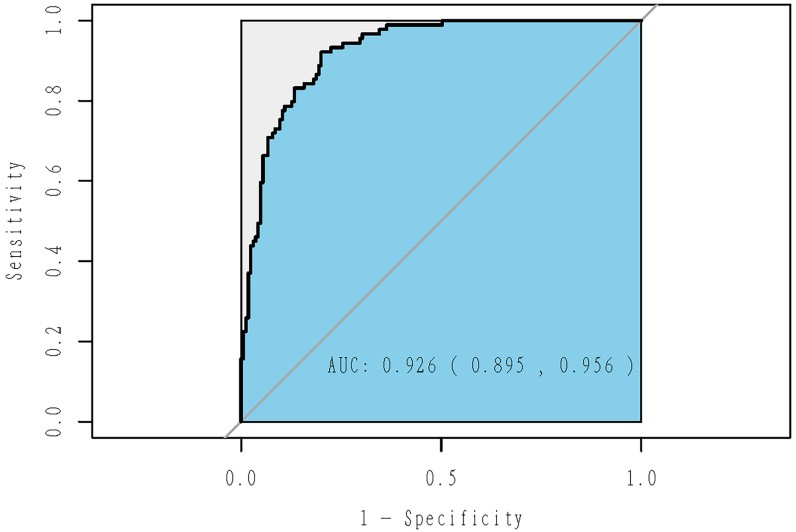

**Figure 4 ROC curve of adaptive Lasso model using the whole sample.** This model explained 48.23% of the deviance, with an AUC (obtained by the bootstrapping) of 0.926 (95% CI [0.895–0.956]; $P = 2.42 \times 10^{-29}$).

**Table 3 Logistic regression results using variables selected by adaptive Lasso (male).**

|  | Coefficient | OR | 95% CI | P |
|---|---|---|---|---|
| Intercept | −129.538 | – | – | 0.003 |
| Ratio of arm to stature | 0.072 | 1.074 | [0.496–2.353] | 0.856 |
| Ratio of arm to leg length A | 0.063 | 1.065 | [0.71–1.596] | 0.760 |
| Torso length | 0.386 | 1.471 | [1.194–1.859] | 0.001 |
| Leg length A to H | 1.45 | 4.261 | [2.007–10.43] | <0.0001 |
| Head circumference | −0.812 | 0.444 | [0.256–0.708] | 0.002 |
| Ratio of waist to chest | −0.426 | 0.653 | [0.488–0.836] | 0.002 |
| Ratio of waist to hip | 0.43 | 1.537 | [1.23–1.997] | <0.0001 |
| Biceps circumference difference | 1.02 | 2.773 | [1.393–6.152] | 0.006 |
| Skeletal muscle mass | 0.176 | 1.192 | [1.001–1.443] | 0.056 |

**Notes:**
Nine out of 23 variables were selected for male swimmers.
OR, odds ratio; CI, confidence interval.
Arm means upper limb length (cm); Leg length A means iliospinale posterior height (cm); Torso length means acromion height—trochanterion height (cm); Leg length H means eristailiaca height (cm); Biceps circumference difference means maximum biceps circumference—minimum biceps circumference (cm).

In stratified analysis by gender, nine out of 23 variables were selected for male swimmers (Table 3), while eight variables for female athletes (Table 4). The model for male explained 46.49% of the deviance, with an AUC of 0.921 (95% CI [0.880–0.963]; $P = 8.82 \times 10^{-17}$; Fig. 5). The model for female explained 54.85% of the deviance, with an AUC of 0.941 (95% CI [0.898–0.984]; $P = 7.67 \times 10^{-15}$; Fig. 6).

## DISCUSSION

In this study, we built a mathematical model using anthropometric factors to identify elite freestyle swimmers (≤400 m) through an adaptive Lasso approach. The adaptive Lasso

**Table 4 Logistic regression results using variables selected by adaptive Lasso (female).**

|  | Coefficient | OR | 95% CI | *P* |
|---|---|---|---|---|
| Intercept | 0.659 | – | - | 0.980 |
| Ratio of arm to stature | −0.065 | 0.937 | [0.359–2.429] | 0.891 |
| Ratio of arm to leg length A | 0.357 | 1.429 | [0.892–2.373] | 0.146 |
| Torso length | 0.332 | 1.393 | [1.058–1.88] | 0.022 |
| Head circumference | −0.918 | 0.399 | [0.201–0.694] | 0.003 |
| Ratio of waist to chest | −0.391 | 0.676 | [0.51–0.853] | 0.002 |
| Ratio of waist to hip | 0.321 | 1.379 | [1.136–1.732] | 0.002 |
| Skeletal muscle mass | 0.512 | 1.669 | [1.239–2.434] | 0.002 |
| Body fat percentage | −0.076 | 0.927 | [0.781–1.075] | 0.344 |

**Notes:**
Eight out of 23 variables were chosen for female athletes.
OR, odds ratio; CI, confidence interval.
Arm means upper limb length (cm); Leg length A means iliospinale posterior height (cm); Torso length means acromion height—trochanterion height (cm).

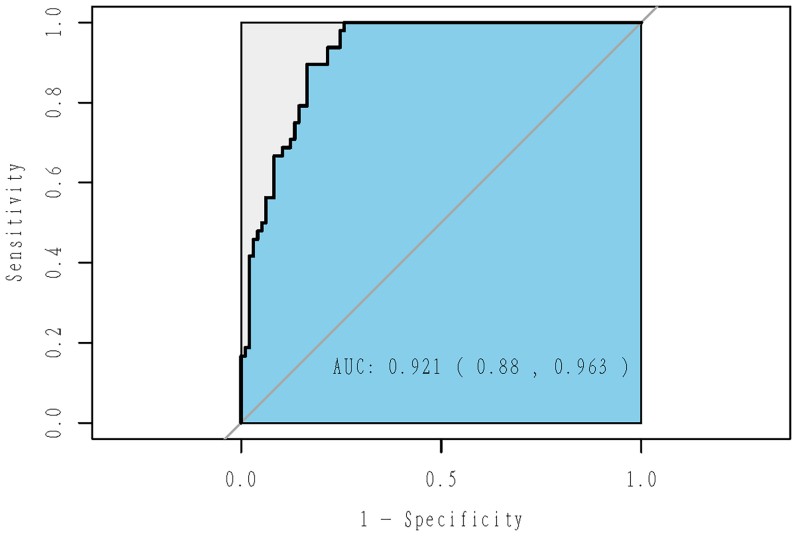

**Figure 5 ROC curve of adaptive Lasso model for male swimmers.** The model for male explained 46.49% of the deviance, with an AUC of 0.921 (95% CI [0.880–0.963]; $P = 8.82 \times 10^{-17}$).

selected 12 variables using the whole sample, and six of them were consistently selected in sex-specific analysis. The models showed satisfactory performance in the identification of elite short-to-medium-distance freestyle swimmers. To the best of our knowledge, this is the first study that adopted the adaptive Lasso approach to identify elite swimmers using anthropometric characteristics.

Fat free mass, which primarily consists of skeletal muscle, was reported to be one of the best predictors of swimming performance (*Lätt et al., 2009a*; *Nevill, Oxford & Duncan, 2015*). For example, freestyle swimming speed could increase by 33.1% with an increase of 1-kg lean body mass (*Nevill, Oxford & Duncan, 2015*). More lean body mass often means more appendicular lean mass as well as more skeletal mass in the limbs, leading to greater

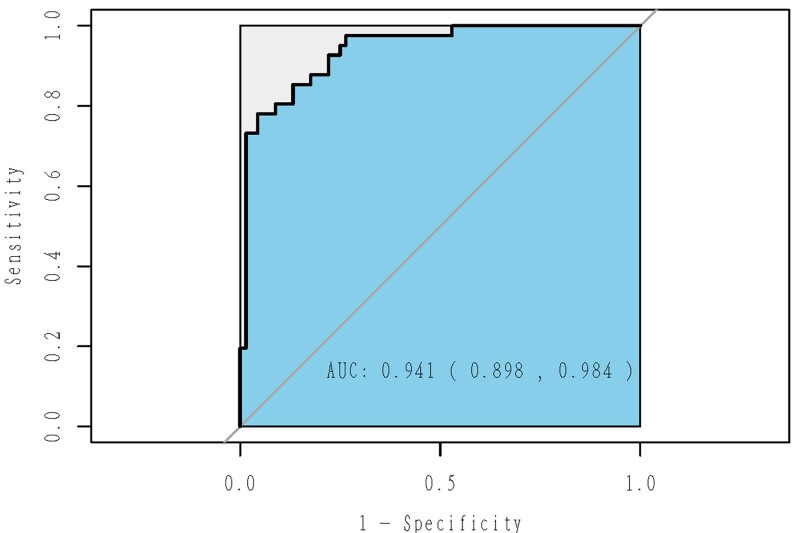

**Figure 6 ROC curve of adaptive Lasso model for female swimmers.** The model for female explained 54.85% of the deviance, with an AUC of 0.941 (95% CI [0.898–0.984]; $P = 7.67 \times 10^{-15}$).

stroke rate and subsequent propulsive force, and finally quicker swimming speed (*Nevill, Oxford & Duncan, 2015*; *Sammoud et al., 2018b*). Consistent with these findings, in our study, skeletal muscle mass was selected as a significant predictor of elite swimmers: with 1-kg increase in skeletal muscle mass, the odds of becoming an elite swimmer increased by 33% (OR = 1.33, 95% CI [1.17–1.54]; $P < 0.0001$) and 67% (OR = 1.67, 95% CI [1.24–2.43]; $P = 0.002$) in the analysis using the whole sample and female swimmers, respectively (Tables 2 and 4). Although skeletal muscle mass was not significantly associated with being an male elite swimmers, it still contributed to the prediction accuracy (OR = 1.19, 95% CI [1.00–1.44]; Table 3). These findings corroborated the importance of muscularity component in performance prediction and talent identification.

The trunk is an essential segment for swimming performance. Long torso was found to positively affect backstroke swimming speed (*Sammoud et al., 2019*). Expanding to the overall competitive performance (*i.e.*, not restricted to one stroke and/or distance), torso length (sitting height) was the only anthropometric variable selected in the multiple lineal regression model in male swimmers, and sitting height explained 34% of the variance in swimming performance (*Saavedra, Escalante & Rodríguez, 2010*). In our study, we found that torso length contributed to the prediction of being an elite swimmer: with 1-cm increase in torso length, the odds of becoming an elite swimmer increased by 34% (OR = 1.34, 95% CI [1.15–1.59]; $P < 0.0001$). Stratification analysis by gender yielded similar results (Tables 3 and 4). These results were further supported by previous findings regarding the advantage of having shorter legs in freestyle swimming, as wave drag could be reduced due to longer torso, leading to increased hull speed and decreased energy consumption (*Nevill, Oxford & Duncan, 2015*; *Papic & Sanders, 2018*; *Sammoud et al., 2019*). Long torso is thus a promising indicator of elite swimmers.
In addition to length of the trunk, horizontal size of the trunk also plays an important role in swimming performance. Biacromial breadth (r = −0.61, P < 0.01) and biiliocristal breadth (r = −0.46, P < 0.01) were significantly correlated with swimming sprint performance in young male swimmers (*Geladas, Nassis & Pavlicevic, 2005*). Furthermore, each 1-cm increase in biacromial breadth and biiliocristal breadth increased 100-m butterfly speed by 48.9% and 29.2%, and 100-m breaststroke speed by 56.5% and 40.3%, respectively (*Sammoud et al., 2018a, 2018b*). In our study, trunk breadth ratio (*i.e.*, bitrochanteric breadth/biacromial breadth) and trunk circumference ratios (*i.e.*, waist circumference/chest circumference, and waist circumference/hip circumference) were selected as significant predictors: in the whole sample, a 1-unit increase in the first two ratios decreased the odds of being an elite swimmer by 18.2% (OR = 0.818, 95% CI [0.698–0.947]; P = 0.009) and 23.2% (OR = 0.768, 95% CI [0.647–0.897]; P = 0.001), respectively; while a 1-unit increase in the last ratio increased the odds by 27% (OR = 1.27, 95% CI [1.107–1.474]; P = 0.001) (Table 2). Similar results for trunk circumference ratios were observed in stratified analysis by gender (Tables 3 and 4). These observations suggested that greater biacromial breadth and chest circumference were associated with higher likelihood of being an elite swimmer, while bitrochanteric breadth and hip circumference showed a reverse association with the likelihood of being an elite swimmer. Our results, combined with the negative association of head circumference, supported that a 'streamlined' body shape was essential to an elite swimmer as it could reduce resistance during the underwater phase (*Sammoud et al., 2018b*). Together, our findings suggested that coaches should take trunk into account as a key anthropometric predictor of being an elite swimmer.

We found different indicators for male and female elite swimmers, implying that there may be differences in anthropometric characteristics that affect swimming performance in male and female swimmers. Previous literature attributed gender differences in swimming performance to differences in somatic features (*Knechtle et al., 2020*). For example, having a higher lean body mass and lower body fat in male swimmers compared to females was found to be significantly associated with shoulder strength and swimming performance (*Avlonitou et al., 1997*). In addition, anthropometric characteristics also contributed to differences in swimming biomechanics (*e.g.*, upper and lower segment coordination, energy cost, and swimming economy) between male and female swimmers. For instance, female swimmers have a smaller body size, which allows them to have lower body torque due to reduced distance between the center of mass and the center of volume, leading to a smaller body cross-sectional area in the water and improved swimming economy (*Seifert, Barbosa & Kjendlie, 2011*).

Early (de)selection and early specialization is a concern for long-term athlete development, while early diversification is considered as a remedy measure for early specialization. Both early age of peak performance of swimmers and the "10-year rule" suggested the need for early specialization of swimmers (*Born et al., 2022*). However, the unique performance environment of swimming limits effectiveness of early diversification practices in transferring performance elements and cross-training from other diversified sports (*Baker, 2003*). Nonetheless, dry-land strength training programs are still important

for swimmers as upper-body strength was found to contribute significantly to the full stroke cycle: with 1-kg increase in the upper-body dry-land strength, the thrust is increased by 0.397 N (95% CI [0.189–0.605]; $P = 0.002$) (*Morais et al., 2020a*).

This study has some limitations. We only considered anthropometric factors in the model, and the included somatic indicators were not comprehensive. Future studies are needed to explore whether adding other factors (*e.g.*, physiological and bioenergetical factors) and other anthropometric factors (*e.g.*, frontal surface area, trunk transverse surface area, maximal rate of change in cross-sectional area, and posterior form gradient of the waist-hip) that indicate actual athletic swimming performance can further improve the performance of the prediction model (*Morais et al., 2011*; *Morais et al., 2020b*; *Papic et al., 2020*). In addition, considering test efficiency, we only used the traditional soft tape to collect circumference features, while several studies reported that new procedures, such as 3D scanners, could provide reliable and accurate data (*Rumbo-Rodríguez et al., 2021*). We only used cross-sectional data in model building. It is possible that changes in anthropometric factors over time may be better indicators for identification of elite swimmers (*Abbott & Collins, 2002*). Future studies utilizing longitudinal data and advanced analytical methods, such as joint modeling (*Wulfsohn & Tsiatis, 1997*) and time-varying effects modeling (*Tan et al., 2012*), might provide better predictive performance. Joint modeling takes both longitudinal follow-up data and time-to-event information into account, leading to more precise estimation and more accurate prediction than modeling the two types of data separately (*Koslovsky et al., 2020*; *Li & Luo, 2019*; *Rizopoulos et al., 2016*), while time-varying effects modeling targets intensive longitudinal data, and can provide dynamic predictions when more longitudinal data are available, making it ideal for identification of elite swimmers. To represent the homogeneity of swimmers as much as possible, event specificity was not strictly controlled for in this study. Future studies are needed to explore anthropometric factors associated with each specific event by recruiting swimmers who participate in the same events after sample size calculation. In addition, our study is of retrospective nature. Prospective studies that align sampling time with the time at which an athlete achieves his/her highest level could provide more robust and accurate prediction. This study was conducted on Chinese swimmers, and it is not known whether our findings can be generalized to swimmers of other ethnicities. More studies with data from other ethnicities are needed to validate our findings.

## CONCLUSIONS

The present study adopted the adaptive Lasso method to select anthropometric factors to establish a model for the prediction of elite short-to-medium-distance freestyle swimmers. Our analysis pinpointed several anthropometric factors that were associated with being an elite swimmer, such as torso length, leg length A to H, head circumference, ratio of waist to hip, and skeletal muscle mass. The models showed satisfactory predictive performance, implying that they might be helpful to coaches in the identification of elite freestyle swimmers in the late adolescence. They also indicated that the adaptive Lasso approach can be applied in talent identification in other sports.

### Funding

This work was supported by the Science and Technology Commission of Shanghai Municipality under Grant (Nos. 16JC1400503, 18DZ1200600, and 22DZ1205100). The funders had no role in study design, data collection and analysis, decision to publish, or preparation of the manuscript.

### Grant Disclosures

The following grant information was disclosed by the authors:
Science and Technology Commission of Shanghai Municipality: 16JC1400503, 18DZ1200600, and 22DZ1205100.

### Competing Interests

The authors declare that they have no competing interests.

### Author Contributions

- Qile Pan conceived and designed the experiments, performed the experiments, analyzed the data, prepared figures and/or tables, authored or reviewed drafts of the article, and approved the final draft.
- Rongxin Zhu performed the experiments, analyzed the data, prepared figures and/or tables, and approved the final draft.
- Jun Qiu conceived and designed the experiments, authored or reviewed drafts of the article, and approved the final draft.
- Guang Cai conceived and designed the experiments, performed the experiments, authored or reviewed drafts of the article, and approved the final draft.

### Human Ethics

The following information was supplied relating to ethical approvals (*i.e.*, approving body and any reference numbers):

The Ethics Committee of the School of Life Sciences of Fudan University granted Ethical approval to carry out the study (No. 497).

### Data Availability

The raw measurements are available in the Supplemental File.

### Supplemental Information

Supplemental information for this article can be found online at http://dx.doi.org/10.7717/peerj.14635#supplemental-information.

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
