# Peer review of "Construction of an anthropometric discriminant model for identification of elite swimmers: an adaptive lasso approach"

_PeerJ, doi:10.7717/peerj.14635_

## Round 0.1 · original submission · Major Revisions

The reviewers agreed on the merit and potential of the article. However, significant revisions should strengthen the clarity and overall quality.

Reviewer 3 has suggested that you cite specific references. You are welcome to add it/them if you believe they are relevant. However, you are not required to include these citations, and if you do not include them, this will not influence my decision.

·

Basic reporting

1. The aim of this study is to establish a predictive model of elite freestyle swimmers in base to anthropometric variables using for this an adaptative Lasso approach. However, in order for this model to be used as a reference in talent detection is necessary theoretical foundation in this topic.
2. Structure conforms to PeerJ Standards.
3. AUC is not defined in his first apparition in the Abstract (l.23).

Experimental design

4. The study design explanation is missing (l. 62), which generates many doubts. For example, to stablish the number of participants is representative. What criteria has been followed? Was any sample size analysis carried out?
5. The participants in the study are swimmers that compete in short to medium distance freestyle (l. 62), Do all swimmers compete in the same distances? How do you establish that an event is medium distance? Are there any differences between swimmers of short distance i.e., 50m and swimmers of longer distances i.e. 400 or 1500m?
It is presented as a study limitation the lack of other factors like the physiological or bionergetical that could improve the prediction model (l. 194). However, I would relevant for this study to differentiate between different kinds of distance specialization of the swimmers. This could help to discuss which anthropometric factors are more relevant for each event.
6. The results obtained by the swimmers are classified in elite and non-elite (l.66-69). Were those results achieved at the same time that the study was developed? If that not the case, it could be more interesting to use the personal time in each event measured in the same moment that the anthropometric data.
7. It this specified that the somatic measurements were taken off-season (l.80). If the aim is to analyzed performance related the variables, would it not be more relevant to take de measurements during the season? Many variables are influenced by the physical condition of the swimmer: e.g., fat free mass, as you reference, can be one of the best predictors of swimming performance (l. 147-150).
8. It is specificized that the model presented can be used us a valid talent identification model (l.158, l.211). However, this study gives us information about a singular predictive model of the performance, making the assumption that this can be used like talent identification isn´t advisable.
There are many factors that will not allow to make this model generalized approach: age is not be accounted to analyzed performance. E.g., it is referenced that muscular component is relevant for talent identification (l.157-158), but of which stages of development are we speaking of? In models of talent detection is paramount to take into account the development stages of the athletes and the development of the physical abilities according to their maduration stage. Related to this topic, I recommend the reading of following works:
• Vaeyens, R., Lenoir, M., Williams, A.m. et al. Talent Identification and Development Programmes in Sport. sports med 38, 703–714 (2008). https://doi.org/10.2165/00007256-200838090-00001
• Armstrong N, McManus AM (eds): The Elite Young Athlete. Med Sport Sci. Basel, Karger, 2011, vol 56, pp 1–22 https://doi.org/10.1159/000320618)
Many of the information gathered in the predictive models can be used for the processes of detection of talents, even more so in a sport like swimming, where physical capacities are especially relevant. However, exposing a model without a solid theoretical scientific foundation associated to each variable can lead to reproduce mistakes as early (des)selection and the early specialization. To expand on this topic:
• Angela Abbott & Dave Collins (2002) To Theoretical and Empirical Analysis of to 'State of the Art' Talent Identification Model, High Ability Studies, 13:2, 157-178, DOI: 10.1080/1359813022000048798.
• Baker, J. (2003). Early specialization in youth sport: To requirement for adult expertise? High ability studies, 14(1), 85-94.

Validity of the findings

9. The procedure hardly achieves that another researcher can reproduce his work, follows the guidelines of a manual of internal training and does not specify in the document as it is the procedure (l.76).
10. Please be more specific on the conclusions. What are the main findings of his study?
11. What are the main practical applications think that could have his model and with which populations could be used?

·

Basic reporting

The analyzed manuscript is well written, with adequate structure and precise references. On the other hand, there are minor scoring structure problems, there is a need for further discussion of certain results and revision of more suitable terms. No hypothesis was formulated, but based on previous studies, with other methodologies, especially focusing on the relationship between drag anthropometry and propulsion, it would be possible to formulate hypotheses.The choice of the LASSO methodology is unprecedented and contributes both to the general development of sport sciences and, specifically, to swimming.

Experimental design

The experimental design is adequate and unprecedented in the swimming area. It is well described and allows replicating the study in other environments and cultures.

Validity of the findings

The results are valid and allow the discussion to be developed in an appropriate and original way.

Additional comments

In the abstract backgroud, please insert "The main of this study was to build..." or something similar.
In the abstract (and along the manuscript) if the number is lesss than 10 and without units, please writte the number in full (nine variables, not 9 variables).
In the abstract: Plase define AUC, not use the abbreviation here.
In the Introduction and along the text: References cannot be separated from the preceding sentence, which they justify, by a period or comma. Dots or commas must come after the references:
not "trunk breadth,(Sammoud et al., 2018a)", but trunk breadth (Sammoud et al., 2018a)"
or
not "factor affecting sport performance.(Barbosa et al., 2010; Saavedra et al., 2010; Ugarkovic et al., 2002) In..."
but
"factor affecting sport performance (Barbosa et al., 2010; Saavedra et al., 2010; Ugarkovic et al., 2002). In..."
Please end the introduction with the objectives of the manuscript
Material & Methods
Lines 63 to 65: do not identify the Universities
Line 77 (and along the paper): It's not "circumference", but "perimeter"
Results
Line 120: Please change "made" to "build"
Discussio
LIne 135: Please insert "mathematical" before "model"
Discussion: The long torso importance is related to the "wave drag" decreasing? A long torso could improve the "hull speed"? Please discuss these possibilities.
Why the variables were not the same for men and women? Please explain and discuss it.

Reviewer 3 ·

Basic reporting

Please see attached file

Experimental design

Please see attached file

Validity of the findings

Please see attached file

Additional comments

Please see attached file

Annotated reviews are not available for download in order to protect the identity of reviewers who chose to remain anonymous.

---

## Round 0.2 · accepted · Accept

The article was improved and can be accepted in its current form.

·

Basic reporting

Thanks for following my comments and suggestions. I consider that the information added to the article improves its quality. In addition, questions that couldn´t be implemented in the article have been answered coherently.

Experimental design

No comment

Validity of the findings

No comment

·

Basic reporting

The authors performed a major review of the original article. The modifications allowed a better understanding of the logic of the study, the results, and the discussion. I understand that the points raised above have been duly answered.

Experimental design

I am satisfied with the changes made.

Validity of the findings

I am satisfied with the changes made.

Additional comments

I am satisfied with the changes made.

Reviewer 3 ·

Basic reporting

The authors improved the quality of the manuscript based on the reviewers' comments.
The manuscript is now more clearer for readers.

Experimental design

Significantly improved.

Validity of the findings

Signifcantly improved.

Additional comments

The authors improved the quality of the manuscript based on the reviewers' comments.
The manuscript is now more clearer for readers.